# Towards Efficient Pre-Trained Language Model via Feature Correlation Distillation

**Kun Huang**[1]    **Xin Guo**[1]    **Meng Wang**[1*]
[1]Ant Group
{hunterkun.hk,darren.wm,bangzhu.gx}@antgroup.com

## Abstract

Knowledge Distillation (KD) has emerged as a promising approach for compressing large Pre-trained Language Models (PLMs). The performance of KD relies on how to effectively formulate and transfer the knowledge from the teacher model to the student model. Prior arts mainly focus on directly aligning output features from the transformer block, which may impose overly strict constraints on the student model's learning process and complicate the training process by introducing extra parameters and computational cost. Moreover, our analysis indicates that the different relations within self-attention, as adopted in other works, involves more computation complexities and can easily be constrained by the number of heads, potentially leading to suboptimal solutions. To address these issues, we propose a novel approach that builds relationships directly from output features. Specifically, we introduce token-level and sequence-level relations concurrently to fully exploit the knowledge from the teacher model. Furthermore, we propose a correlation-based distillation loss to alleviate the exact match properties inherent in traditional KL divergence or MSE loss functions. Our method, dubbed FCD, presents a simple yet effective method to compress various architectures (BERT, RoBERTa, and GPT) and model sizes (base-size and large-size). Extensive experimental results demonstrate that our distilled, smaller language models significantly surpass existing KD methods across various NLP tasks.

## 1  Introduction

Past few years have witnessed a rapid development of pre-trained language models (PLMs) thanks to their effectiveness across a wide range of natural language processing tasks. Pre-trained language models, such as BERT Devlin et al. [2018], RoBERTa Liu et al. [2019], and GPT-2 Radford et al. [2019], learn contextualized text representations by predicting words given their context using large scale text corpora, and can be fine-tuned with additional task-specific layers to adapt to downstream tasks. However, the excellent capability for various NLP tasks comes at demaning huge resources and large memory footprints. For example, the BERT$_{\text{BASE}}$ model contains about 110M parameters and 12 Transformer Vaswani et al. [2017] layers, which prevents these transformer-based models from being finetuned and deployed on resource-constrained devices and real-time applications. Recent studies Kovaleva et al. [2019], Voita et al. [2019] indicate that redundancy exists in the original PLMs. Therefore, a series of attempts Chung et al. [2020], Wu et al. [2020], Wang et al. [2020c], Gordon et al. [2020a], Tang et al. [2019], Aguilar et al. [2019] have been made to review the techniques for effective compression of the pre-trained heavy transformers without compromising the performance, of which knowledge distillation is considered to be a practical paradigm, Typically, knowledge distillation techniques aims at effectively transferring the dark knowledge embedded in a large teacher network to boost the performance of the smaller student network during training. Once trained, this compact

---

*Corresponding author.

37th Conference on Neural Information Processing Systems (NeurIPS 2023).

student network can be directly deployed in real-life applications without introducing extra inference time or structure modifications. The essence of knowledge distillation relies on how to formulate and transfer the knowledge from teacher to student. The classic logit-based distillation Hinton et al. [2015] directly mimic the final prediction outputs between the teacher and student via Kullback–Leibler (KL) divergence, which only brings limited performance gain to the student. Besides this vanilla knowledge distillation, many other works Du et al. [2020], Heo et al. [2019], Tian et al. [2019] also try to make use of intermediate representations of the pre-trained teacher transformer. The intermediate layers contain more embedding, supplement richer features, thus allow the student transformer to acquire more information in addition to outputs. Jiao et al. [2019] proposed TinyBERT, which distill the information between multiple intermediate features, including the embedding, self-attention matrices and hidden states of the teacher and student networks via the mean squared error (MSE). However, this usually need the adaption layers to align the mismatching embedding dimensions. Such gap makes it hard for the student to mimic the teacher's feature directly and induces additional training cost as a consequence. MINILM Wang et al. [2020b] employs self-attention heatmaps and and value relations via the KL-divergence loss to deeply mimic teacher's self-attention modules. It leads to a restriction that the number of attention heads of student model has to be the same as its teacher. To solve this problem, MINILMv2 Wang et al. [2020a] first concatenate and then split self-attention vectors of different attention heads according to the desired number of relation heads, which involves more computation of queries, keys, and values in self-attention and consequently leads to suboptimal performance.

Motivated by these observations, we aim to directly model feature relationships between the teacher and student models. In a manner similar to the self-attention mechanism, we first model token relations using the output features of the teacher and student models. This token-level relationship has the capacity to capture long-term dependencies between input tokens and highlight critical tokens essential for linguistic comprehension. While the token-level relation is intuitive, it reflect only one aspect of the feature relationships. We further exploit another important aspect of feature relations, the sample-level, to capture the semantic relationship across a batch of samples, an aspect that has been overlooked in previous works. In this way, both types of relations are combined to complement each other to bring more fine-grained feature knowledge. Therefore, the student model is expected to have superior performance compared to that trained stand-alone. Thanks to the same shape of feature relations between student and teacher, our method offers increased flexibility with respect to the embedding dimension and the number of attention heads. Moreover, we propose a correlation-based loss function to replace the KL divergence and MSE used in traditional KD methods. Specifically, we employ the pearson linear correlation as a novel loss function, relaxing the exact match property typically associated with KL divergence and MSE. The overview of the proposed method is illustrated by Figure 1. To sum up, our main contributions are outlined as follows:

- We directly model feature relationships between teacher and student models, which jointly leveraging token-level and sample-level relations to distill knowledge for the first time.
- We propose a correlation-based loss function using Pearson linear correlation, and theoretically explain that it offers a more flexible alternative to traditional KL divergence and MSE.
- Extensive experiments are conducted with popular variants of PLMs, including BERT, RoBERTa, and GPT on GLUE datasets, and our proposal consistently performs better than existing methods.

## 2 Related Works

### 2.1 Pretrained Language Models

Pretrained language models are pretrained on large amounts of text corpus, and then fine-tuned on task-specfic dataset. BERT Devlin et al. [2018] proposes to use a masked language modeling (MLM) objective to pretrain a deep bidirectional Transformer encoder. RoBERTa Liu et al. [2019] achieves strong performance by training longer steps using large batch size and more text data. Besides those encoder-based models, GPT-2 Radford et al. [2019] is a decoder-based model designed for uni-directional, left-to-right text processing. It predicts the next word in a sequence given the preceding words, which allows it to generate coherent and contextually relevant text. In additon to knowledge distillation, the compression of pretrained language models have been widely explored, ranging from

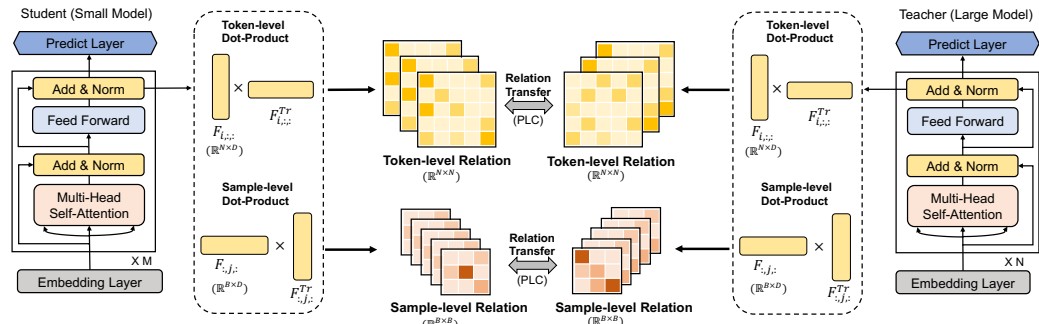

Figure 1: Overview of the proposed method Feature Correlation Distillation (FCD). To demonstrate the effectiveness of FCD. We introduce the token-level relationship and sample-level relationship to distill the knowledge from teacher to student. The correlation loss based on pearson linear correlation is used to capture the relationship between the teacher and the student (best viewed in color).

unstructured pruning Gordon et al. [2020b], Guo et al. [2019], attetnion head pruning Michel et al. [2019]; to layer factorization Lan et al. [2019], quantization Zhang et al. [2020], Bai et al. [2020] and dynamic width/dpeth inference Hou et al. [2020]. However, some of the techniques like weight pruning (irregular sparsity) and quantization typically require complex piplines and can not lead to inference speedup and run-time memory saving directly without dedicated hardware/libraries (e.g. for sparse or low-bit computing operation). By contrast, knowledge distillation has been found to be a simple and much effective model compression technique that allows a relatively simple model to perform tasks almost as accurately as a complex model. Moreover, it can be combined with other compression techniques (i.e., where the student model is a smaller, quantized, or pruned version of the teacher model) to further compress the pre-trained language models.

## 2.2 Knowledge Distillation

Knowledge Distillation (KD) is a process that transferring knowledge from a large teacher model to a small student model. It was first proposed by Hinton et al. [2015] and then how to effectively transfer more knowledge has been explored by many subsequent works Romero et al. [2014], Ahn et al. [2019], Park et al. [2019], Tian et al. [2019], Tung and Mori [2019]. The intermediate layers contain much richer representation, thus allow the student transformer to acquire more information in addition to outputs. Tang et al. [2019] distill fine-tuned BERT into an extremely small bidirectional LSTM. Turc et al. [2019] initialize the student with a small pre-trained LM during task-specific distillation. Sun et al. [2019a] introduce the hidden states from every k layers of the teacher to perform knowledge distillation layer-to-layer. Aguilar et al. [2019] further introduce the knowledge of self-attention distributions and propose progressive and stacked distillation methods. Task-specific distillation requires to first fine-tune the large pre-trained LMs on downstream tasks and then perform knowledge transfer. The procedure of fine-tuning large pre-trained LMs is costly and time-consuming, especially for large datasets. MiniBERT Tsai et al. [2019] uses the soft target distributions for masked language modeling predictions to guide the training of the multilingual student model and shows its effectiveness on sequence labeling tasks. DistillBERT Sanh et al. [2019] uses the soft label and embedding outputs of the teacher to train the student. TinyBERT Jiao et al. [2019] and MOBILE-BERT Sun et al. [2019b] further introduce self-attention distributions and hidden states to train the student. For example, MOBILE-BERT employs inverted bottleneck and bottleneck modules for teacher and student to make their hidden dimensions the same. TinyBERT uses a uniform-strategy to map the layers of teacher and student when they have different number of layers, and a linear matrix is introduced to transform the student hidden states to have the same dimensions as the teacher. However, the presence of those extra modules not only adds burden on network complexity but also complicates the training procedure.

## 3 Method

In this section, we first describe the transformer architectures and define the distillation target in Section 3.1. Then we introduce the two proposed types of feature relationships: token-level relation

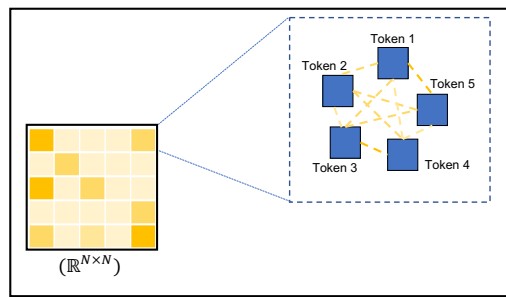
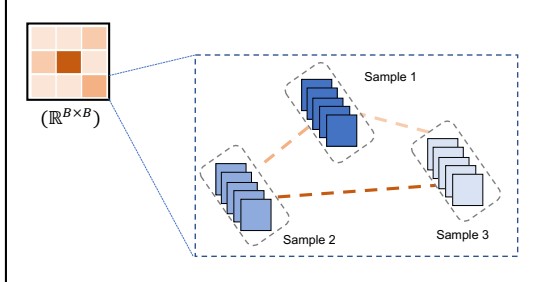

**Figure 2:** Our method proposes to maintain the token-level and sample-level relations between student and teacher. Token-level relation: relation between the tokens within each sample of teacher and student. Sample-level relation: relation between the samples on a specfic token.

in Section 3.2 and sample-level relation in Section 3.2. At last, we elaborate on the correlation-based loss in Section 3.3 and provide a theoretical analysis in Section 3.4.

### 3.1 Preliminaries

The majority of pre-trained language models are based on Transformer architectures, which are composed of a stack of Transformer blocks. We first tokenize the input sample into a sequence of tokens and pack them together with special tokens such as [SEP] and [CLS]. These tokens are then projected into token embeddings and fed into transformer blocks. Each Transformer block consists of a Multi-Head Attention (MHA) and a Feed-Forward network (FFN). Layer normalization (LN) Ba et al. [2016] and residual connection He et al. [2016] are integrated around each of these two sub-blocks. Suppose a teacher model $T$ and a student model $S$, the model takes the feature $\mathcal{F}_{l-1}$ of the $l$-th Transformer block as input, In multi-head attention, heads are computed in parallel to get the final output, which can be formulated as:

$$\begin{aligned} \mathcal{A}_l &= \text{Attn}(\mathcal{F}_{l-1}, Q_l, K_l) \\ \mathcal{H}_l &= \sum_H \mathcal{A}_l V_l \end{aligned} \quad (1)$$

$\mathcal{A}_l \in \mathbb{R}^{H \times N \times N}$ is the attention matrix, where $H$ denotes the number of heads and $N$ the sequence length of the input. It is calculated as scaled dot-product between $Q_l$ and $K_l$ and then apply softmax operation on the each column of matrix $\mathcal{A}_l$. The final multi-head attention output $\mathcal{H}_l$ is calculated as a weighted sum of values $V_l$. Suppose the two linear layers in FFN are parameterized by $W_1$, $b_1$ and $W_2$, $b_2$, the output of FFN can be formulated as:

$$\mathcal{F}_l = \text{GeLU}(\mathcal{H}_l W_1 + b_1) W_2 + b_2 \quad (2)$$

We term $\mathcal{F}_l \in \mathbb{R}^{N \times D}$ as the output feature of the $l$-th Transformer block, where $D$ denotes the dimension of hidden features. Some works directly adopted $\mathcal{F}_l$ as the distillation target. However, the embedding dimensions $D_S$ and $D_T$ of student and teacher are typically different. Previous works Romero et al. [2014], Yim et al. [2017], Heo et al. [2019] overcome this obstacle by building certain adaptation modules between hidden layers of the teacher and student models. However, these adaptation modules, with random initialization or special non-parameter transformation Srinivas and Fleuret [2018], Komodakis and Zagoruyko [2017] would potentially disturb training process, because it introduces extra parameters and computational cost (including weights, gradients and optimizer states) Pudipeddi et al. [2020]. Moreover, the teacher and student models usually have different number of heads, i.e., $H_T \neq H_S$. To address these challenges, we aim to model feature relationships using $\mathcal{F}_l$ to overcome the aforementioned issues. The details of this process are provided in the following sections.

### 3.2 Distillation with feature relationships

**Token-level Relation**    To mitigate the negative influences of magnitude differences, we first normalize the block features $\mathcal{F}$ of both the teacher and the student, denoted as $\hat{\mathcal{F}} = \text{Norm}(\mathcal{F})$. Common

normalization methods include $\ell_2$, softmax and layer normalization. In our implementation, we choose $\ell_2$ normalization as its implementation, as it consistently outperforms the other methods in our experiments. Subsequently, with the normalized feature, we compute the token-level relation matrix. Specifically, for a specific sample $i$, we define the relation matrix between tokens for each sample as follows:

$$\mathcal{R}_t(\mathcal{F}) = \frac{\mathcal{F}_{i,:,:}}{||\mathcal{F}_{i,:,:}||_2} \cdot \left(\frac{\mathcal{F}_{i,:,:}}{||\mathcal{F}_{i,:,:}||_2}\right)^{Tr} \tag{3}$$

Here, $Tr$ denotes transposition for the feature. For each given sample, both the student's and teacher's token-level relation matrices share the same dimensions of $\mathcal{R}_t \in \mathbb{R}^{N \times N}$. Each matrix effectively serves as a relevance map, revealing the influence of each token in relation to others within the same sequence (see the left part of Figure 2). By doing so, this inner token-level relationship is adept at capturing long-term dependencies between tokens, emphasizing those which are crucial for a comprehensive understanding of the linguistic context. Consider tasks such as text tagging or named entity recognition, where each token must be assigned a label based on its role within the sentence. In such cases, the token-level relation matrix can serve as an effective guide, assisting the student model to deliver enhanced performance.

**Sample-level Relation**  In addition to the token-level relation, which captures the relationships among different tokens within each sample, the relationships across multiple samples for each token also provides informative knowledge for the distillation process. As such, we aim to distill this sample-level relation as well to enhance performance. Similarly, we compute the relation matrix at the sample level as follows:

$$\mathcal{R}_s(\mathcal{F}) = \frac{\mathcal{F}_{:,j,:}}{||\mathcal{F}_{:,j,:}||_2} \cdot \left(\frac{\mathcal{F}_{:,j,:}}{||\mathcal{F}_{:,j,:}||_2}\right)^{Tr} \tag{4}$$

In this case, for a specific token, both the student's and teacher's sample-level relation matrices, denoted as $\mathcal{R}_s$, share the same dimensions of $\mathbb{R}^{B \times B}$. These matrices describe the relationships between different samples, as illustrated in the right part of Figure 2. They enable the capture of semantic relationships across a batch of samples, which is particularly essential for tasks like text classification and summarization. In these tasks, recognizing the sample-level relationship can be instrumental in making accurate predictions. It allows the model to discern the similarities and differences between samples, rather than viewing each sample in isolation. For the sample-level relation modeling, instead of directly comparing words based on their positions across sentences, we transform each sentence into a unified high-dimensional space using the same network. Within this space, token features at the same positions from distinct sequences become comparable. This approach forms the foundation of our sample-level relation modeling and serves as a valuable guide for the student model, enhancing its performance in tasks necessitating a deep understanding of inter-sample relationships.

**Computation Complexity Analysis**  The token-level and sample-level relation maps, in our proposed method, require computational complexities of $2BN^2D$ and $2B^2ND$, respectively. The associated memory space required for these computations is $BN^2 + B^2N$. In contrast, MiniLM Wang et al. [2020b] utilizes self-attention and value-value attention, results in total computational complexities and memory space requirements of $4BN^2D$ and $2BN^2$, respectively. MiniLMv2 Wang et al. [2020a] employs a different attention mechanism, resulting in total computational complexities and memory space requirements of $18BN^2D$ and $9BN^2$, respectively. For a concrete example, consider the $\text{BERT}_{\text{BASE}}$ model, where $B$ is set to 32, $N$ is 128, and $D$ is 768. Our proposed method yields computational complexities and memory space requirements of 0.6GFLOPs and 0.6MB, respectively. In comparison, MiniLM has computational complexities and memory space requirements of 3.2GFLOPs and 2MB, respectively, while for MiniLMv2, these values rise to 14.5GFLOPs and 9.4MB, respectively. Thus, our method demonstrates significant efficiency in terms of computational complexity and memory space requirement, providing a more resource-efficient option for knowledge distillation.

### 3.3 Distillation with pearson linear correlation

In addition the distillation target, the design of the distillation loss plays a crucial role in transferring the knowledge from the teacher model to the student model. A general distillation loss can be

expressed as:

$$\mathcal{L} = \Gamma(\phi_T(\mathcal{F}_T), \phi_S(\mathcal{F}_S)) \tag{5}$$

Here, $\phi_T$ and $\phi_S$ denote the feature transformations for the teacher and student models, respectively. These transformations convert raw features into a form that is more conducive to knowledge transfer. In previous works, $\phi$ often takes the form of an adaptation layer that aligns the embedding dimensions between the teacher and student models. However, in our method, as discussed in Section 3.2, $\phi$ is used to denote the token-level and sample-level relation matrix. $\Gamma$ is a distance function measuring the similarity between student and teacher features. The most commonly used functions include KL divergence and the mean squared error. However, these functions exhibit an 'exact match' property, which means the loss reaches the minimal if and only if the features of student and teacher are all identical. This requirement could impose overly strict constraints on the student, particularly when there is a large discrepancy between the teacher and student. To alleviate these constraints, we introduce the Pearson correlation coefficient as a more flexible alternative, which is used to measure the strength of the linear relationship between two variables and remains invariant under positive linear transformations. The basic form of Pearson correlation coefficient between $X$ and $Y$ follows the Pearson index can be computed as follows:

$$\rho(X, Y) := \frac{\sum (X_i - \mu_X)(Y_i - \mu_Y)}{\sqrt{\sum (X_i - \mu_X)^2}\sqrt{\sum (Y_i - \mu_Y)^2}} \tag{6}$$

where $\mu_X, \mu_Y$ denote the mean value of the variable $X, Y$ respectively. The Pearson correlation coefficient, denoted by $\rho$, ranges between -1 and 1. Therefore, $1 - \rho$ always falls between 0 and 2, making it suitable to be used as a loss function. We call our final formulation of the distance function Pearson linear correlation (PLC), which is defined as follows:

$$\Gamma(X, Y) := 1 - \rho(X, Y). \tag{7}$$

In this way, we shift the focus from attempting to exactly replicate the teacher's features to preserving and learning the relational information between the teacher's and student's features. This shift effectively relaxes the 'exact match' requirement inherent in conventional Knowledge Distillation (KD) methods. Thus, the distillation loss in our approach comprises two types of losses: token-level relation loss and sample-level relation loss. The token-level relation loss quantifies the discrepancy between the token-level relation matrices of the student and teacher models. It is defined as the average PLC between the student's and teacher's token-level relations across all samples in the batch:

$$\mathcal{L}_t := \frac{1}{B} \sum_{i=1}^{B} \Gamma(\mathcal{R}_t(\mathcal{F}_S), \mathcal{R}_t(\mathcal{F}_T)) \tag{8}$$

The sample-level relation loss quantifies the discrepancy between the sample-level relation matrices of the student and teacher models. It is defined as the average PLC between the student's and teacher's sample-level relations across all tokens in the sequence.

$$\mathcal{L}_s := \frac{1}{N} \sum_{j=1}^{N} \Gamma(\mathcal{R}_s(\mathcal{F}_S), \mathcal{R}_s(\mathcal{F}_T)) \tag{9}$$

The intrinsic sensitivity of the PLC to outliers necessitates normalization of features prior to calculating the distillation loss, thus ensuring a stable training process. Given these two components, the overall training loss of our proposed method consists of the task loss, token-level relation loss, and sample-level relation loss, which can be formulated as follows:

$$\mathcal{L} = \mathcal{L}_g + \alpha \mathcal{L}_t + \beta \mathcal{L}_s \tag{10}$$

Here, $\mathcal{L}_g$ denotes the task training loss, while $\alpha$, $\beta$ are weighting factors used to balance the task training loss and the relation losses.

### 3.4 Theoretical Analysis

As discussed in Section 3.3, different distance functions like Kullback-Leibler (KL) divergence and Mean Squared Error (MSE) are commonly used to measure the similarity between the teacher and the student models. However, these functions have an 'exact match' property, which means that the distance is zero if and only if the student and teacher features are exactly the same. This

| Model | #Params | Speedup | SST-2 | MNLI-m | QNLI | QQP | RTE | SST-B | MRPC | CoLA | AVG |
|---|---|---|---|---|---|---|---|---|---|---|---|
| BERT$_{BASE}$ | 110M | ×1.0 | 93.4 | 84.5 | 91.5 | 72.3 | 66.8 | 85.2 | 88.3 | 52.8 | 79.3 |
| BERT$_{SMALL6}$ | 66M | ×2.0 | 90.7 | 81.2 | 87.9 | 69.4 | 64.3 | 79.8 | 83.7 | 41.4 | 74.8 |
| BERT-PKD$_6$ | 66M | ×2.0 | 92.0 | 81.5 | 89.0 | 70.7 | 65.5 | 81.6 | 85.0 | 43.5 | 76.1 |
| DistilBERT$_6$ | 66M | ×2.0 | 92.5 | 82.6 | 88.9 | 70.1 | 58.4 | 81.3 | 86.9 | 49.0 | 76.2 |
| TinyBERT$_6$ | 66M | ×2.0 | 92.1 | 82.8 | 89.8 | 71.2 | 70.0 | 83.9 | 88.0 | 51.1 | 78.6 |
| MINILM$_6$ | 66M | ×2.0 | 92.0 | 83.0 | 91.1 | 71.4 | 70.8 | 84.2 | 88.5 | 49.2 | 78.8 |
| MINILMv2$_6$ | 66M | ×2.0 | 92.4 | 83.4 | 90.0 | 71.5 | 71.3 | 84.5 | 88.6 | 51.8 | 79.2 |
| **Ours** | 66M | ×2.0 | **92.8** | **83.8** | **91.3** | **72.0** | **71.7** | **84.8** | **89.1** | **52.0** | **79.6** |
| RoBERTa$_{BASE}$ | 125M | ×1.0 | 95.3 | 87.2 | 93.2 | 73.8 | 72.7 | 88.4 | 90.1 | 62.0 | 82.8 |
| RoBERTa$_{SMALL6}$ | 82M | ×2.0 | 92.3 | 83.1 | 90.4 | 72.1 | 68.4 | 86.8 | 87.5 | 54.1 | 79.3 |
| MINILMv2$_6$ | 82M | ×2.0 | 93.5 | 84.3 | 91.6 | 72.8 | 72.1 | 87.5 | 88.2 | 57.8 | 81.0 |
| **Ours** | 82M | ×2.0 | **93.8** | **85.6** | **92.0** | **73.5** | **72.5** | **88.3** | **89.6** | **60.3** | **81.9** |
| DistilGPT2 | 82M | ×2.3 | 90.7 | 81.6 | 87.9 | 66.8 | 68.3 | 79.6 | 87.9 | 39.4 | 75.3 |
| **Ours** | 82M | ×2.3 | **92.0** | **83.4** | **88.5** | **70.6** | **70.2** | **81.6** | **88.4** | **42.3** | **77.1** |

Table 1: Results of the proposed method on the test sets of GLUE. We use the metric of Matthews correlation for CoLA, Pearson-Spearman correlation for STS-B, and accuracy for other datasets. Following previous works Sun et al. [2019a], we also report the average score of these eight tasks (the "AVG" column). The speedup is in terms of the BERT$_{BASE}$ and RoBERTa$_{BASE}$ inference time and evaluated on a single GPU with a single input of 64 or 128 length. The fine-tuning results are an average of 4 runs.

can be written as $\Gamma(X, Y) = 0$ when $X = Y$. Unlike KL divergence or MSE, PLC is invariant under positive linear transformations. This means that even if a positive linear transformation is applied to one or both of the features, the correlation coefficient remains the same. In other words, $\Gamma(X, Y) = 0$ if $Y = \alpha X + \beta$, where $\alpha > 0$ and $\beta$ are constants. This property makes PLC a more flexible choice for knowledge distillation, as it allows the student model to learn from the teacher model in a less restrictive way. The detailed mathematical justification for this property is provided in the supplementary materials.

We next delve into the relationship between the Pearson linear correlation (PLC), Kullback-Leibler (KL) divergence and Mean Squared Error (MSE). For KL divergence, the normalized features are transformed into a probability distribution, and then we minimize the discrepancy between the softened probabilities of the teacher and student models.

$$\mathcal{L}_{KL}(\hat{\mathcal{F}}_T, \hat{\mathcal{F}}_S) = \tau^2 \sum_m \phi_t(\hat{\mathcal{F}}_T; \tau) \log \frac{\phi_t(\hat{\mathcal{F}}_T; \tau)}{\phi_s(\hat{\mathcal{F}}_S; \tau)} \tag{11}$$

Here, $\tau$ denotes a temperature parameter used to adjust the softness degree in the distributions. Assuming the value of $\tau$ is significantly large compared to the magnitude of the normalized features, and $\hat{\mathcal{F}}_S, \hat{\mathcal{F}}_T$ are drawn from a standard normal distribution, we can derive the gradient of $\mathcal{L}_{KL}$ with respect to the normalized feature $\hat{\mathcal{F}}_S^i$ as follows:

$$\frac{\partial \mathcal{L}_{KL}}{\partial \hat{\mathcal{F}}_S} \approx \frac{1}{M} \left( \hat{\mathcal{F}}_S - \hat{\mathcal{F}}_T \right) = \frac{\partial \mathcal{L}_{MSE}}{\partial \hat{\mathcal{F}}_S} \tag{12}$$

Further, given that $\frac{1}{m-1} \sum_i \hat{\mathcal{F}}_S^2 = 1$ and $\frac{1}{m-1} \sum_i \hat{\mathcal{F}}_T^2 = 1$, we can reformulate the MSE loss as follows:

$$\mathcal{L}_{MSE}(\hat{\mathcal{F}}_S, \hat{\mathcal{F}}_T) = \frac{1}{2M} \sum (\hat{\mathcal{F}}_S - \hat{\mathcal{F}}_T)^2 \approx 1 - \rho(\mathcal{F}_S, \mathcal{F}_T) \tag{13}$$

This equivalence clarifies the intrinsic connection between KL divergence, MSE, and PLC of normalized feature distributions. The detailed derivations and empirical results to support this claim are provided in the supplementary material.

## 4 Experiments

### 4.1 Distillation Setup

We evaluate the efficacy of the proposed FCD on 8 out of 9 tasks from the General Language Understanding Evaluation (GLUE) Wang et al. [2019] benchmark, which consists of 2 single-sample

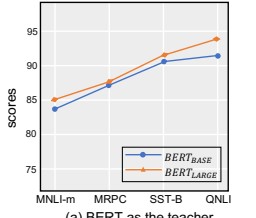 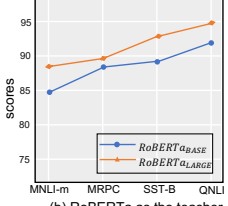 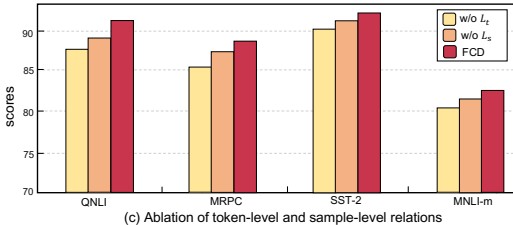

Figure 3: Comparison of distillation results of student models using BASE-size and LARGE-size teachers (BERT and RoBERTa) shown in the left part. The right part shows ablation of token-level and sample-level relations.

(CoLA and SST-2) and 5 sample-pair (MRPC, QQP, MNLI, QNLI and RTE) classification tasks, and 1 regression task (STS-B). Following previous worksSun et al. [2019a], we use the same metrics as the GLUE benchmark for evaluation. In order to verify the effectiveness and robustness of our method, we distill teacher models with different architectures, model sizes. Concretely, we consider encode-based models including $BERT_{BASE}$, $BERT_{LARGE}$, $RoBERTa_{BASE}$ and $RoBERTa_{LARGE}$ to ensure a fair comparison with of a wide range of prior works. Moreover, we also explore the compression of decoder-based models by employing our distillation method to improve fine-tuning of DistilGPT2 Wolf et al. [2020], which is rarely investigated in most previous works.

## 4.2   Implementation Details

The process of distilling pretrained Language Models (LM) generally comprises two stages: task-agnostic distillation and task-specific distillation. Task-agnostic distillation involves a pre-training process on a large-scale dataset. However, this stage can be costly and time-consuming, particularly for scenarios with limited computational resources. In contrast, task-specfic LM distillation proves to be effective and considerably more economical compared to pre-training. As an increasing number of pretrained LM models are becoming publicly available through resources such as the HuggingFace Transformers library Wolf et al. [2020], directly leveraging these models can save substantial time and computational resources compared to training from scratch. Given these considerations, our focus in this work is on task-specific knowledge distillation. However, it is important to note that our method is not limited to task-specific distillation and can be readily applied in task-agnostic scenarios as well. Specifically, we first fine-tune the pretrained teacher models on a specific task. Following this, the corresponding student model is initialized with the teacher model using the LayerDropping method Sajjad et al. [2020]. Subsequently, we perform distillation with FCD. We employ a grid search algorithm on the development set to tune the hyper-parameters. Specifically, we trained the student model for 3, 5 and 10 epochs, using a batch size of 32. The learning rates we experimented included $2e-5$, $1e-5$ and $5e-6$. For the CoLA task, we extended the training steps to 25 epochs. The parameters $\alpha$ and $\beta$ from the distillation loss are tuned from $\{0.1, 0.2, 0.4, 1\}$, a choice guided by maintaining the different components of the loss in the same order of magnitude. We adopt a cosine decay schedule with a warm-up phase of 5 epochs and utilize the AdamW optimizer with a weight decay of 0.5. The maximum sequence length is set to 64 for single-sample tasks, and 128 for sequence pair tasks.

## 4.3   Main Results

We start by comparing our proposed method with several KD baselines, including DistilBERT Sanh et al. [2019], TinyBERT Jiao et al. [2019], BERT-PKD Sun et al. [2019a], MiniLM Wang et al. [2020b], MiniLMv2 Wang et al. [2020a]. Similar to previous studies, we distill a 12-layer base model into a 6-layer student model with only about 60% parameters and 2x inference speedup. In order to evaluate the impact of knowledge distillation, we also report the results of $BERT_{SMALL6}$ and $RoBERTa_{SMALL6}$. These smaller models are obtained using the LayerDropping method Sajjad et al. [2020], wherein the strategy of dropping the top layer has been demonstrated to be a strong baseline. Consequently, we drop the top 6 layers of the base model and fine-tune it without using knowledge distillation. For fair comparisons, we fine-tune the released models and evaluate the result on the test set of GLUE without resorting to data augmentation strategy Jiao et al. [2019]. The results

Table 2: Comparison of using different distillation loss functions.

| Method | MNLI-m | QQP | RTE | Average |
|---|---|---|---|---|
| FCD (KL) | 83.3 | 71.2 | 70.7 | 75.1 |
| FCD (MSE) | 83.6 | 71.6 | 71.2 | 75.5 |
| FCD (Pearson) | **83.8** | **72.0** | **71.7** | **75.8** |

Table 3: Distillation results with different layer selecting schemes.

| Scheme | layers | MNLI-m | CoLA |
|---|---|---|---|
| Top | {7,8,9,10,11,12} | 82.9 | 50.7 |
| Uniform | {2,4,6,8,10,12} | 83.8 | 52.0 |
| Bottom | {1,2,3,4,5,6} | 83.3 | 51.5 |

from these 6-layer student models are summarized in Table 1. The top group of models denotes the uncased version of base-size BERT used as the teacher model.Notably, our model outperforms all compared models by a large margin. Similar trends are observed in the middle group, where base-size RoBERTa is utilized as the teacher model. Our model surpasses the MiniLMv2 by 0.4% accuracy on RTE, 0.5% F1 on MRPC, and 0.3% Spearman correlation on STS-B. Furthermore, we conduct experiments on a decode-base model DistilGPT2. In this setting, we employ the GPT-2 model with 12 layers and 768 hidden size as the teacher model. Noatably, our proposed method outperforms the original KD method by an average of 1%, , underlining the efficacy of our approach in decoder-based model. As illustrated in the left part of Figure 3, student models that are distilled from large-size teacher models achieve further improvements. Moreover, this performance gain increases with the capacity of teacher models, thereby demonstrating the effectiveness of our proposed method across different sizes of pretrained Transformer models.

## 4.4 Ablation Studies

**Effect of different components**    In this study, we introduce two distinct types of relations: token-level and sample-level relations. To verify the effectiveness of each, we conduct experiments using $\mathcal{L}_t$ and $\mathcal{L}_s$ to investigate their respective influences on the student model. As depicted in the right part of Figure 3, each component within the distillation loss independently contributes to the enhancement of the final performance. Moreover, a further boost in performance is observed when these components are combined. We noticed that the performance degradation on SST-2 is more substantial compared to other tasks without token-level relation. We speculate that the token-level relation is particularly important for this single sentence binary classification task.

**Effect of different distillation loss functions**    Here, we compare our proposed Pearson Linear Correlation (PLC) with the Mean Squared Error (MSE) and Kullback–Leibler (KL) divergence, which are widely-used loss functions. To ensure a fair comparison, we tune the distillation loss weight for both MSE and KL. The comparative results across three tasks are presented in Table 2, which demonstrates that adopting Feature Correlation Distillation (FCD) with PLC consistently yields higher performance compared to the FCD combined with MSE and KL. This indicates that the more flexible Pearson correlation might serve as a more suitable metric for measuring relations within the FCD.

**Effect of different layer selection strategies**    Apart from the type of knowledge used for distillation, the selection of layers significantly affects the overall performance of distillation. We study the impact of three distinct layer selection strategies: uniform, top, and bottom, and compare their respective performances. Specifically, we utilize BERT$_{\text{BASE}}$ as the teacher model and a $6 \times 768$ model as the student model. The number of selected layers is set to 6. The results are reported in Table 3. For BERT$_{\text{BASE}}$, using the uniform layer selection strategy yields superior performance compared to the other strategies. This finding highlights the crucial role of both the head and tail layers of student models in the distillation process.

## 5    Conclusion

In this paper, we introduce Feature Correlation Distillation (FCD), a novel and effective method for distilling large Transformer-based Pretrained Language Models (PLMs). Our approach simultaneously models both token-level and sample-level relations derived from the features of the Transformer block. Moreover, we propose a correlation-based distillation loss to enhance the performance of the model distillation process. We also provide a theoretical interpretation of our proposed Pearson

linear correlation formulation, offering a deeper understanding of its underlying operation and implications. Through extensive experiments on the GLUE tasks, our distilled smaller language models consistently outperformed existing knowledge distillation methods across a variety of architectures while significantly reducing both the model size and inference time. With its simplicity and strong performance, we hope our approach can serve as a solid baseline for future research.

## Acknowledgements

This research was supported by Ant Group. We are grateful to the five anonymous reviewers for their insightful suggestions and comments, which significantly enhanced the quality of this paper.

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

# 6 Appendix

## 6.1 Implementation of FCD

The implementation details of FCD are presented in Figure 4. Utilizing the output features from both the student and teacher models, denoted as $F_s$ and $F_t$ respectively, our method ensures ease of implementation.

```python
import torch.nn as nn
from torch.nn import functional as F

def cosine_similarity(x, y, eps=1e-8):
    return (x * y).sum(1) / (x.norm(dim=1) * y.norm(dim=1) + eps)

def pearson_correlation(x, y, eps=1e-8):
    return cosine_similarity(x - x.mean(-1).unsqueeze(-1),
                             y - y.mean(-1).unsqueeze(-1), eps)

def plc_loss(x, y):
    return 1 - pearson_correlation(x, y).mean()

def token_relation_loss(y_s, y_t):
    R_s = y_s.bmm(y_s.transpose(-1, -2))
    R_t = y_t.bmm(y_t.transpose(-1, -2))
    R_s = R_s.view(R_s.size(0), -1)
    R_t = R_t.view(R_t.size(0), -1)
    loss = plc_loss(R_s, R_t)
    return loss

def sample_relation_loss(y_s, y_t):
    y_s = y_s.permute(1, 0, 2)
    y_t = y_t.permute(1, 0, 2)
    loss = token_relation_loss(y_s, y_t)
    return loss

class FCDLoss(nn.Module):
    def __init__(self, alpha, beta):
        super(FCDLoss, self).__init__()
        self.alpha = alpha
        self.beta = beta

    def forward(self, F_s, F_t):
        F_s = F.normalize(F_s, dim=-1)
        F_t = F.normalize(F_t, dim=-1)
        loss_token = token_relation_loss(F_s, F_t)
        loss_sample = sample_relation_loss(F_s, F_t)
        kd_loss = self.alpha * loss_token + self.beta * loss_sample
        return kd_loss
```

Figure 4: The PyTorch implementation of FCD.

## 6.2 Proofs

**Invariance of Pearson's Correlation under Positive Linear Transformation.** Let's consider two random variables $X$ and $Y$. A positive linear transformation on $X$ and $Y$ can be formulated as $X' = aX + b$ and $Y' = cY + d$, where $a \times c > 0$ and $b$, $d$ are arbitrary constants. Applying these transformations to the means of $X$ and $Y$, we derive $\mu_{X'} = a\mu_X + b$ and $\mu_{Y'} = c\mu_Y + d$. By substituting the transformed variables and their corresponding means into Equation 6:

$$
\begin{aligned}
\rho'(X,Y) &= \frac{\sum((aX_i + b) - (a\mu_X + b))((cY_i + d) - (c\mu_Y + d))}{\sqrt{\sum((aX_i + b) - (a\mu_X + b))^2}\sqrt{\sum((cY_i + d) - (c\mu_Y + d))^2}} \\
&= \frac{\sum(a(X_i - \mu_X))(c(Y_i - \mu_Y))}{(a\sqrt{\sum(X_i - \mu_X)^2})(a\sqrt{\sum(Y_i - \mu_Y)^2})} = \rho(X,Y)
\end{aligned}
\tag{14}
$$

**Relationship among PLC, KL divergence and MSE** In Equation 11, $\phi(\cdot)$ represents the softmax function, while $\tau$ denotes a temperature parameter controlling the softness of the distributions.

$$
\frac{\partial \mathcal{L}_{KL}}{\partial \hat{\mathcal{F}}_S^i} = \tau \left( \phi_s(\tau) - \phi_t(\tau) \right) = \tau \left( \frac{\exp(\hat{\mathcal{F}}_S^i/\tau)}{\sum_{j=1}^m \exp(\hat{\mathcal{F}}_S^j/\tau)} - \frac{\exp(\hat{\mathcal{F}}_T^i/\tau)}{\sum_{j=1}^m \exp(\hat{\mathcal{F}}_T^j/\tau)} \right),
\tag{15}
$$

Assuming $\tau$ is significantly large compared to the magnitude of the normalized features and both $\hat{\mathcal{F}}_S^i$ and $\hat{\mathcal{F}}_T^i$ are drawn from a standard normal distribution. In this case, the term $\hat{\mathcal{F}}^i/\tau$ becomes quite small, allowing us to approximate $\exp(\hat{\mathcal{F}}^i/\tau)$ as $1 + \hat{\mathcal{F}}^i/\tau$. This simplification leads to an



Figure 5: Visualization of the token-level relational features under different normalization functions. (a) Pre-Norm. (b) $\ell_2$-Norm. (c) Layer-Norm. (d) Softmax-Norm.

approximation of the gradient in Equation 15:

$$\frac{\partial \mathcal{L}_{KL}}{\partial \hat{\mathcal{F}}_S^i} \approx \tau \left( \frac{1 + \hat{\mathcal{F}}_S^i/\tau}{M + \sum_{j=1} \hat{\mathcal{F}}_S^j/\tau} - \frac{1 + \hat{\mathcal{F}}_T^i/\tau}{M + \sum_j \hat{\mathcal{F}}_T^j/\tau} \right) \tag{16}$$

Given that the sums $\sum_j \hat{\mathcal{F}}_S^j$ and $\sum_j \hat{\mathcal{F}}_T^j$ are both zero, Equation 16 simplifies further to:

$$\frac{\partial \mathcal{L}_{KL}}{\partial \hat{\mathcal{F}}_S^i} = \frac{1}{M}(\hat{\mathcal{F}}_S - \hat{\mathcal{F}}_T) = \frac{\partial \mathcal{L}_{MSE}}{\partial \hat{\mathcal{F}}_S^i} \tag{17}$$

Moreover, considering $\frac{1}{m-1}\sum_i \hat{\mathcal{F}}_S^2 = 1$ and $\frac{1}{m-1}\sum_i \hat{\mathcal{F}}_T^2 = 1$, we can reformulate the MSE as follows:

$$\begin{aligned}
\mathcal{L}_{MSE}(\hat{\mathcal{F}}_S, \hat{\mathcal{F}}_T) &= \frac{1}{2M}\sum(\hat{\mathcal{F}}_S - \hat{\mathcal{F}}_T)^2 \\
&= \frac{1}{2M}\left( (2M-2) - 2\sum_{i=1}^{m} \hat{\mathcal{F}}_S\hat{\mathcal{F}}_T \right) \\
&= \frac{2M-2}{2M}(1 - \rho(\mathcal{F}_S, \mathcal{F}_T)) \approx \mathcal{L}_{PLC}(\mathcal{F}_S, \mathcal{F}_T)
\end{aligned} \tag{18}$$

Thus, we demonstrate that minimizing KL divergence between normalized features under a high-temperature limit is equivalent to minimizing the MSE between normalized ones, which is in turn equivalent to maximizing the PLC between the original features.

### 6.3 More Experiments Results

**Effect of different Normalization and Loss Functions** Section 3.4 in the main text clarifies the intrinsic relationship between KL divergence, MSE, and PLC. However, the assumption that normalized features follow a Gaussian distribution may not invariably be valid. To investigate the performance of varying normalization and loss functions, we conducted a series of experiments, setting the temperature $\tau$ to 10 when utilizing KL divergence as the loss function. Table 4 demonstrates that the $\ell2$ norm consistently outperforms other normalization functions. Minimizing KL divergence between layer-normalized features in the high-temperature limit can yield results comparable to MSE and PLC. To further underscore the benefits of $\ell2$ normalization, we provide a visualization of the pre-normalized and post-normalized token-level relational features in Figure 5. In contrast to $\ell2$ normalization, other functions often produce a wider range with larger values, suggesting that directly imitating these normalized features could introduce significant noise, potentially leading to subpar results.

**Results on SQuAD v1.1 and v2.0.** To further demonstrate FCD's effectiveness, we applied it to the question-answering tasks of SQuAD v1.1 Rajpurkar et al. [2016] and SQuAD v2.0 Rajpurkar et al. [2018]. We framed these tasks as sequence labeling problems, predicting the likelihood of each token being the start or end of an answer span. We employed the F1 metric for both versions of SQuAD. BERT$_{BASE}$ was used as the teacher model, and a $6 \times 768$ model served as the student model. The results, presented in Table 5, indicate that FCD can enhance the student model's performance on both tasks.

Table 4: Results of the loss function combined with different normalization mechanism.

| Scheme | MNLI-m | QQP | CoLA | Average |
|---|---|---|---|---|
| LayerNorm + KL | 83.4 | 71.5 | 51.4 | 68.8 |
| LayerNorm + MSE | 83.6 | 71.8 | 51.5 | 69.0 |
| PreNorm + PLC | 83.3 | 71.2 | 51.7 | 68.7 |
| Softmax + PLC | 83.4 | 71.7 | 51.5 | 68.9 |
| $\ell_2$ + PLC | 83.8 | 72.0 | 52.0 | 69.3 |

Table 5: Results of baselines and FCD on question answering tasks.

| Method | SQuAD 1.1 | SQuAD 2.0 |
|---|---|---|
| $BERT_{BASE}$ | 88.7 | 78.8 |
| $DistilBERT_6$ | 86.2 | 69.5 |
| $TinyBERT_6$ | 87.5 | 77.7 |
| **Ours** | 88.2 | 78.4 |

## 6.4 Discussion

**Limitations.** While FCD demonstrates consistent performance improvements across diverse Transformer-based models, its effectiveness may be less pronounced on other architectures such as Recurrent Neural Networks (RNNs). The feature relationships in RNNs are not as explicit as in Transformers, potentially limiting the applicability and impact of FCD.

**Societal impacts.** The extensive computational resources required to evaluate our proposed method could significantly contribute to carbon emissions, thereby raising sustainability concerns. However, the objective of our approach is to enhance the efficiency of lightweight models through knowledge distillation. This enhancement could ultimately replace heavier models in production settings, resulting in substantial energy savings. Thus, the thorough validation of FCD's efficacy is a necessary trade-off to ensure its potential benefits.

