# OpenReview forum: "Towards Efficient Pre-Trained Language Model via Feature Correlation Distillation"
_NeurIPS.cc/2023/Conference — NeurIPS 2023 poster_

### Official Review · Reviewer_R5h8 · 2023-06-20

**Soundness:** 2 fair
**Presentation:** 3 good
**Contribution:** 3 good
**Rating:** 4
**Confidence:** 3

**Summary:**

This paper introduces a new technique for compression the PLM. Within the context of knowledge distillation, the authors introduce a new type of relation distillation, which focuses on modeling relations between token features at both the token-level and sentence-level. These relations are then utilized as an objective function for the teacher-student distillation process. Additionally, the authors propose a correlation-based distillation loss as a replacement for the previous distillation loss function that relied on exact match as the target. Experimental results show that FCD achieves significant improvements over the previous methods.

**Strengths:**

1. A clear motivation for the method: resolving the mismatch in size during alignment between the teacher and the student, such as the number of heads or the dimension of the hidden state. The authors propose modeling both token-level and sample-level relationships, which remains consistent between the teacher and the student models, regardless of their model design. This approach can help to minimize the impact of alignment mismatch.
2. The method offers a simple and effective solution for distillation, supported by proof that demonstrates the relationship between PLC, KLD, and MSE.
3. The presentation of method is clear and easily comprehensible.
4. It is remarkable to observe that FCD, a task-specific distillation approach, surpasses the performance of task-agnostic KD with pre-training, like TinyBERT and MiniLMv2.

**Weaknesses:**

1. The idea of sample-level relation loss appears unreasonable to me. The approach of comparing the j-th token in each sentence and using the dot-product similarity to represent sentence-level relationship seems unreasonable. It raises doubts about whether the relationships between tokens in the same position, but in different sentences, truly reflect the relationships between the sentences and are directly comparable. For instance, consider the following example:
S1: For simplicity and clarity, our paper focuses...
S2: For your safety, please ensure that you are wearing protective...
If we focus on the first token, they are the same, yet the sentences have completely different meanings.

2. I have some doubts regarding the experimental results and would appreciate further clarification from the authors to determine if any potential weaknesses exist. Please refer to questions 1, 2, and 3.

**Questions:**

1. Upon reviewing the results in Table 1, I noticed that none of the reported results on the test set on GLUE benchmark, including BERT_base, are consistent with those presented in their original paper. Some results are higher than the previous paper, while others are lower. This raises the question of whether all the baseline results were reproduced or if there were modifications to the student structure or different settings employed. Could you clarify why there are changes to the baselines results?
2. The baseline "BERT_SMALL6" involves dropping some layers and fine-tuning on the downstream datasets without knowledge distillation (KD), which can be interpreted as the L_g loss in Eq.10. However, I noticed a significant discrepancy between the reported SST-2 result of 90.7% and the result shown in Figure 3.c's ablation experiments, where using only L_g on SST-2 yields approximately 86%. Could you explain the reason behind this substantial gap between the two results?
3. The motivation for FCD is to address the size misalignment problem; however, in the conducted experiments, both the teacher and student models maintain the same structure, except for dropping certain layers. Consequently, there doesn't appear to be a misalignment issue in dimension and the number of attention aheads. Could you provide some experiments on the student model will less attention head and also reduced dimension?
4. Can you provide with some explanation why the sample-level relation loss can aid in distillation?
5. Have you attempted to build FCD upon released distilled models with pre-training, such as TinyBERT (e.g., TinyBERT_General_
*L_*D)? It will be interesting to see combining these two techniques together to boost the performance of the distilled model.

---

> ### Author Rebuttal · Authors · 2023-08-07
>
> Thanks for your valuable comments and efforts in reviewing our paper. We address your comments and questions in the following content.
>
> > Weakness 1: The concept of sample-level relation loss seems unreasonable. Comparing the j-th token in each sentence to determine sentence-level relationships is questionable. For example, sentences "S1" and "S2" both start with the same token but convey different meanings.
>
> Response to W1: Indeed, the example you provided highlights that individual token similarity may not always reflect the holistic meaning of sentences. However, our proposed sample-level relation loss, as described in Eq.9, is not solely based on one isolated token comparison. Instead, it focuses on an aggregate assessment across all tokens in a sequence. To put it simply, our approach isn't just about direct token-to-token comparison but rather the interaction and aggregation of these tokens across an entire sentence.
> Consider the following customer review examples:
>
> S1: "The camera quality is outstanding..."
>
> S2: "The battery life is inadequate..."
>
> S3: "The display clarity is excellent..."
>
> If we assess token positions across these sentences, we observe patterns associated with product attributes (e.g., tokens in the third position) and sentiment (e.g., tokens in the fifth position). This suggests that there's a relationship not only in the individual tokens but in the context they create.  Thus, we define sentence-level relationships with token-wise relation matrices across all positions and model them in a nuanced and meaningful manner.
>
> > Question 1: In Table 1, the results for the GLUE benchmark, including BERT_base, differ from the original paper. Were the baseline results reproduced as is, or were there changes to the student structure or settings? Please explain the discrepancies.
>
> Response to Q1: Thank you for noting the discrepancies in Table 1. We used public pre-trained models for BERT_base, DistilBERT, and TinyBERT. For MiniLM v1 and v2, we re-implemented them due to differences in their publicly available structures and the absence of test set results. All models were fine-tuned using consistent settings. The variations from original papers might be due to training randomness and specific training details, like the omission of data augmentation used in the TinyBert.
>
> > Question 2: The baseline "BERT_SMALL6" involves dropping some layers and fine-tuning on the downstream datasets without KD, which can be interpreted as the L_g loss in Eq.10. However, I noticed a significant discrepancy between the reported SST-2 result of 90.7% and the result shown in Figure 3.c's ablation experiments, where using only L_g on SST-2 yields approximately 86%. Could you explain the reason behind this substantial gap between the two results?
>
> Response to Q2: We appreciate your careful attention to this discrepancy in our paper. The difference arises from the configurations in Figure 3.c where we applied both L_g and L_s, in contrast to BERT_SMALL6 which only utilized L_g. Upon revisiting our setup, we identified that the beta weight factor was mistakenly set too high, causing an imbalance between the losses L_s and L_g.This misconfiguration resulted in the observed performance gap. After correcting this and re-running our experiments, we achieved an average SST-2 score of 91.9% over four runs. We will amend Figure 3.c in the revised version to reflect this.
>
> > Question 3: Could you provide some experiments on the student model will less attention head and also reduced dimension?
>
> Response to Q3: Thank you for your valuable suggestion. We conducted experiments on student model with reduced attention heads and dimensions. Specifically  we set student models to be 6-layer, 6-head, with a hidden size of 384. Here are the results:
>
> | Model | MNLI-m | SST-2 |
> | ------ | ----- | --------|
> | (Teacher) BERT_Base   | 84.5 | 93.4
> | TinyBert_6H_6L_384    | 79.8 | 88.2
> | MiniLMv2_6H_6L_384    | 81.2 | 89.6
> | FCD_6H_6L_384   | 81.8 | 90.3
>
> FCD's consistent outperformance highlights its capability in alleviating this misalignment problem, which aligns with the motivation of our approach. While the performance of these compact models doesn't reach our original paper's results, potentially due to lack of pre-trained initialization, integrating strategies like attention head pruning could further optimize results. We will incorporate these results in the updated version.
>
> > Question 4: Can you provide with some explanation why the sample-level relation loss can aid in distillation?
>
> Response to Q4: Please see our response to W1. Let us know if it does not answer your question.
>
> > Question 5: Have you attempted to build FCD upon released distilled models with pre-training, such as TinyBERT (e.g., TinyBERT_General_ *L_*D)? It will be interesting to see combining these two techniques together to boost the performance of the distilled model.
>
> Response to Q5: Thank you for posing this insightful question.  We explored the idea by building FCD upon pre-trained TinyBERT models and subsequently fine-tuned them on downstream tasks without data augmentation. Here's a summary of the results:
>
> | Model | RTE | SST-B |
> | ----------- | ----------- | ----------- |
> | (Teacher) BertBase   | 66.8 | 85.2
> | TinyBert_6L_768    | 70.0 | 83.9
> | TinyBert_4L_312    | 64.1 | 80.4
> | (FCD) TinyBert_6L_768   | 71.9 | 85.4
> | (FCD) TinyBert_4L_312   | 67.3 | 81.1
>
> Our experiment shows that FCD combined with general distillation not only outperforms the regular TinyBERT model on RTE and SST-B tasks but also those reported in our original paper, which suggests that initializing the student model with general distillation can provide a robust foundation for further fine-tuning with our FCD method. This observation aligns well with the findings presented in other recent work [1].
>
> References:
>
> [1] Lu, Chengqiang, et al. "Knowledge Distillation of Transformer-based Language Models Revisited." arXiv preprint arXiv:2206.14366 (2022).

---

> > ### Comment · Reviewer_R5h8 · 2023-08-17
> >
> > Thanks for your detailed response.
> >
> > I still feel that comparing the same positions between different samples as a way to establish sample-level relations is not quite reasonable. The examples you provided share the same structure, allowing for the comparability of words at the same positions. However, differences in sentence structure, syntax, and semantics can lead to a lack of direct comparability between words at the same positions across sentences. As such, the method of comparing words at the same positions across different samples as a means of establishing sample-level relations could be constrained.
> >
> > Regarding the aggregation of sentence-level information, a common approach in KD involves utilizing the [CLS] representation as the aggregator. This offers computational efficiencies with a complexity of only B^2D, which is lower than B^2ND. Do you have results comparing this one with your methods?

---

> > > ### Author Response · Authors · 2023-08-18
> > >
> > > > Q1: I still feel that comparing the same positions between different samples as a way to establish sample-level relations is not quite reasonable. The examples you provided share the same structure, allowing for the comparability of words at the same positions. However, differences in sentence structure, syntax, and semantics can lead to a lack of direct comparability between words at the same positions across sentences. As such, the method of comparing words at the same positions across different samples as a means of establishing sample-level relations could be constrained.
> > >
> > > Response to Q1: We agree with the potential concerns of comparing tokens at the same positions across varied samples, especially when considering the inherent differences in sentence structure, syntax, and semantics.
> > >
> > > To provide a clearer perspective: while our examples might give the impression of a direct token-to-token comparison, in fact, the term "token" in our paper refers to high-dimensional representations of words situated within a D-dimensional embedding space. It's vital to underline that there isn't a straightforward one-to-one correspondence between input words and their feature positions.
> > >
> > > These high-dimensional representations capture more than just the standalone meaning of a word. They also  integrate its surrounding context and interactions with other words in the sentence. Within this expanded feature space, these embeddings can discern subtle relationships across different samples, transcending mere surface-level token comparisons. Our empirical results showcased in Section 4.4 further validate the efficacy of these high-dimensional, sample-level feature relationships in enhancing the student model's distillation process.
> > >
> > > Moreover, we believe that exploring techniques like dimensionality reduction may offer a clearer depiction of the intricacies inherent in these high-dimensional relation maps. We truly appreciate your insightful feedback, which will certainly guide our efforts in refining and enriching our paper.
> > >
> > >
> > > > Q2: Regarding the aggregation of sentence-level information, a common approach in KD involves utilizing the [CLS] representation as the aggregator. This offers computational efficiencies with a complexity of only B^2D, which is lower than B^2ND. Do you have results comparing this one with your methods?
> > >
> > > Response to Q2: Thank you for drawing attention to the prevalent use of the [CLS] representation in Knowledge Distillation (KD) as an aggregator for sentence-level information. You're right; it's a widely favored method in the community, given its computational efficiencies. In our research, we ventured into this approach. To leverage the [CLS] representation, we extracted the first position from the sample-level relation matrix of size N x B x B, referencing R_s[0, :, :] as per Eq.4. In this setting, we exclusively applied the sample-level relation loss L_s to evaluate the impact of the [CLS] representation. The results are presented in the table below:
> > >
> > > | Model | MRPC | SST-2 |
> > > | ----------- | ----------- | ----------- |
> > > | (Teacher) BertBase   | 88.3 | 93.4
> > > | [CLS] token only   | 84.3 | 91.4
> > > | Ours   | 84.7 | 91.9
> > >
> > > While the [CLS]-only method offers certain computational advantages, our method underscores the significance of interaction and the aggregation of all tokens within a sentence. As our results indicate, our approach potentially enhances distillation performance. It's worth noting that although focusing solely on sample-level relation maps reduces computational complexity, the overall savings in resources are not substantial. This is mainly because token-level relation maps, which tend to consume more computational resources, overshadow the savings gained from the sample-level relation maps, sometimes even at the expense of some performance trade-offs.

---

> > > > ### Comment · Reviewer_R5h8 · 2023-08-21
> > > >
> > > > Thanks for your detailed response.
> > > >
> > > > > This is mainly because token-level relation maps, which tend to consume more computational resources, overshadow the savings gained from the sample-level relation maps, sometimes even at the expense of some performance trade-offs.
> > > >
> > > > Have you calculated what percentage the calculation of these two relation maps (sample-level and token level) occupy during the entire forward computation process, e.g., calculating the FOLPs and the memory consumption? I'm curious about how much speed and memory improvement it could bring to the MiniLM and MiniLMv2 if we take the whole training process into account.

---

> > > > > ### Author Response · Authors · 2023-08-21
> > > > >
> > > > > We sincerely thank you for your efforts in reviewing our responses.
> > > > >
> > > > > > Question1: Have you calculated what percentage the calculation of these two relation maps (sample-level and token level) occupy during the entire forward computation process, e.g., calculating the FOLPs and the memory consumption? I'm curious about how much speed and memory improvement it could bring to the MiniLM and MiniLMv2 if we take the whole training process into account.
> > > > >
> > > > > Response to Q1: As detailed in the "Computation Complexity Analysis" subsection of Section 3.2 in our original paper, our proposed method for computing token-level and sample-level relation maps demands computational complexities of $2BN^2D$ and $2B^2ND$, respectively. The memory requirement for these calculations amounts to $BN^2 + B^2N$. When comparing the computational complexities between the token-level and the sample-level relation maps, the ratio is determined as $N/B$. Given the experimental setting where $B$ = 32 and $N$ = 128, this complexity ratio between token-level and sample-level relation maps is 4. This suggests that the token-level relation maps tend to utilize more computational resources.
> > > > >
> > > > > Regarding MiniLM and MiniLMv2, while we have theoretically assessed the computational complexities and memory demands of the token-level and sample-level relation maps (as illustrated in the aforementioned subsection), it's worth noting that this analysis primarily addresses the computation of the relation maps. The actual overall efficiency, both in terms of speed and memory during the training process, would be intricately tied to specific hardware configurations and hyper-parameters employed.
> > > > > We are currently considering to delve deeper into this in the revised edition of our paper.

---

> > > > > > ### Comment · Reviewer_R5h8 · 2023-08-21
> > > > > >
> > > > > > Thanks for your detailed response. My primary concern is still the reasonability of the sample-wise relation map. I understand that it could improve the performance, but I still feel a bit weird comparing tokens at the same position in different sentences. Besides, another concern (not the primary concern) is that the authors show in the paper that one of the benefits of the relation map is its low complexity. However, since it's only part of the computation and I cannot figure out how much it could benefit the whole training process.
> > > > > >
> > > > > > Thus, I would maintain my score.

---

> > > > > > > ### Author Response · Authors · 2023-08-21
> > > > > > >
> > > > > > > > but I still feel a bit weird comparing tokens at the same position in different sentences.
> > > > > > >
> > > > > > > Thank you for your feedback. To address your main concern, we'd like to highlight two essential aspects of our method:
> > > > > > >
> > > > > > > **Clarifying Sentence Comparisons**: We do not directly compare words based on their positions across different sentences. Instead, each sentence is transformed into a unified high-dimensional representation through the same network. This representation's shape is NxD, where N is sequence length and D is token feature dimension. In this high-dimensional space, token features at the same position from different sequences become comparable, underpinning the basic rationale behind our sample-level relation modeling.
> > > > > > >
> > > > > > > **Rationalizing Position-wise Computation**: Within this uniform high-dimensional embedding space, our computation of the sample-level relationship consists of two steps: a) We firstly apply position-wise operator for individual tokens and b) then aggregate for the entire sequence. To be specific, we compute a correlation matrix for tokens at each position within a batch, resulting in a BxB matrix. We then concatenate these matrices to form a NxBxB matrix,  representing the sample-level relationships. The above computation schema is akin to the calculation of cosine similarity between vectors, where a position-wise product and a summation are executed in tandem. Thus, we believe that our sample-level relation modeling is both logical and effective.
> > > > > > >
> > > > > > > > However, since it's only part of the computation and I cannot figure out how much it could benefit the whole training process.
> > > > > > >
> > > > > > > Thank you for pointing that out. The focus of our paper is the computation of the relation matrix. As such, our computational complexity analysis is primarily centered around this. Based on that, our method's efficiency stands out when compared to other approaches such as MiniLM v2, while achieving better performance. We acknowledge the importance of providing a holistic perspective on potential training time and memory consumption savings. To address this, we are considering the inclusion of  empirical data on these aspects in the revised version of our paper.

---

### Official Review · Reviewer_kWDa · 2023-07-04

**Soundness:** 3 good
**Presentation:** 3 good
**Contribution:** 3 good
**Rating:** 7
**Confidence:** 3

**Summary:**

This paper proposes Feature Correlation Distillation (FCD), an approach for distilling pre-trained transformers. FCD involves a two-part distillation loss: (1) token-level and (2) sample-level, which helps to eliminate dependence on matching dimensionality/architectural details between the student and teacher and improves efficiency by constant factors. The paper also proposes replacing the KL-divergence or MSE based objective with a pearson correlation owing to its invariance to positive linear transformations. Experiments were run on GLUE with bert, roberta and distilgpt2, including comparing KL, MSE and correlation loss objectives and ablating the token and sample level losses.

**Strengths:**

The paper proposes an interesting approach to distillation that relies on model internals while allowing flexibility in architectural choices. It proposes a more efficient distillation procedure which can be useful in resource constrained settings. The paper also makes an effort to empirically justify the chosen loss types and objectives via ablations.

**Weaknesses:**

GLUE tasks can have a large amount of variance due to randomness and due to small test sets, single-run scores may not be reliable (see e.g. https://arxiv.org/pdf/1904.09286.pdf: Section 4.2 note on variance, https://arxiv.org/abs/2010.06595 on small test sets). So, at the very least, averaging across random seeds and including errors bars in Figure 3 is important to understand if the results are robust. Since these experiments provide the only evidence for the claims and since the claims are largely empirical, adding robustness to the experiments seems crucial.

Nits:
The citation formatting appears to be off: try using \citep and \citet with natbib.



**Questions:**

Would it be possible to expand the set of experiments to more tasks? Providing more varied empirical evidence for the approach could make the paper stronger.

---

> ### Author Rebuttal · Authors · 2023-08-07
>
> Thanks for your valuable comments. We address your comments and questions in the following content.
>
> > Weakness 1: GLUE tasks can have a large amount of variance due to randomness and due to small test sets, single-run scores may not be reliable (see e.g. https://arxiv.org/pdf/1904.09286.pdf: Section 4.2 note on variance, https://arxiv.org/abs/2010.06595 on small test sets). So, at the very least, averaging across random seeds and including errors bars in Figure 3 is important to understand if the results are robust. Since these experiments provide the only evidence for the claims and since the claims are largely empirical, adding robustness to the experiments seems crucial. Nits: The citation formatting appears to be off: try using \citep and \citet with natbib.
>
> Response to W1: Thank you for your insightful feedback. We will address your concerns in two parts:
>
> Robustness in Experiments: We absolutely agree with with the points raised about the inherent variance in GLUE tasks and the potential implications for the reliability of single-run scores. All of our current results are already averaged across four different runs (as indicated in Table 1), and we acknowledge that this might not be sufficiently clear in Figure 3. In addition, in our revised paper, we will augment Figure 3 with error bars that capture the variability across these runs, providing a more comprehensive view of the robustness of our results.
>
> Citation Formatting: We're grateful for the feedback on our citation formatting. We'll make the necessary adjustments and ensure the use of \citep and \citet with natbib for consistent and standard citation representation in the updated version of the paper.
>
> > Question 1: Would it be possible to expand the set of experiments to more tasks? Providing more varied empirical evidence for the approach could make the paper stronger.
>
> Response to Q1: Thank you for your valuable suggestion. Broadening the range of tasks evaluated is indeed beneficial for solidifying our empirical evidence. In addition to the tasks mentioned in the main paper, we have extended our experiments to SQuAD 1.1 and 2.0, which have been available in our supplementary materials. Moreover, based on suggestions from reviewer XAJW, we've tested our approach on the WikiText-103 benchmark for generation task. Our model showcased an improved perplexity of 19.8 against distilgpt2's 21.1. This finding underscores the versatility of our methodology, even beyond classification tasks. We're open to expand our empirical evaluation further, based on the tasks recommended by the reviewers, to cement our approach's validity across diverse NLP applications. Your guidance in this regard is invaluable.
>
> | Model | SQuAD 1.1 | SQuAD 2.0 |
> | ------ | ------- | ------ |
> | (Teacher) BERT_Base   | 88.7 | 78.8
> | DistilBERT   | 86.2 | 69.5
> | TinyBERT  | 87.5 | 77.7
> | Ours  | 88.2 | 78.4
>
> | Model | WikiText-103
> | ------ | ------- |
> | (Teacher) GPT2   | 16.3
> | distilgpt2   | 21.1
> | Ours   | 19.8

---

> > ### Comment · Reviewer_kWDa · 2023-08-14
> > **Thanks for the response**
> >
> > Thanks for providing details! Reliability of experiments was my biggest gripe in the initial review which has been addressed in the response. I've raised my score to reflect this!

---

> > > ### Author Response · Authors · 2023-08-16
> > >
> > > Dear Reviewer kWDa:
> > >
> > > Thanks for your feedback!
> > >
> > > Best,
> > >
> > > Paper3038 Authors

---

### Official Review · Reviewer_XAJW · 2023-07-05

**Soundness:** 2 fair
**Presentation:** 3 good
**Contribution:** 2 fair
**Rating:** 5
**Confidence:** 3

**Summary:**

This paper proposes a method for compressing pre-trained language models (PLMs) based on transformer architectures using feature correlation distillation (FCD). FCD models both token-level and sample-level relations between the teacher and student models, and uses a correlation-based loss function to relax the exact match constraint of traditional knowledge distillation methods. FCD achieves strong performance on the the GLUE benchmark with the teacher of the BERT and RoBERTa.

**Strengths:**

A novel perspective to introduce sample-level information to help distillation. The design of learning objective is also interesting.

**Weaknesses:**

The token-level method is very similar to the attention-based method of Minilm. What is the essential difference? Also, this paper selects the last hidden before the prediction head as the feature, but is this really appropriate? Because the last hidden is too close to the output, a lot of information will be lost. Previous work (e.g., kNN-LM) mentioned that using the feature before the final layer’s FFN might be better.

Although the sample-level idea sounds interesting, it doesn’t seem to make much sense. I’m not very clear why the sample-level information in a batch would be useful. If this is really useful, then not only in the distillation setting, but also in general training, sample-level information should be used (such as batch-norm methods). Otherwise, it is not very convincing to say that sample-level information is only useful in distillation.

The results of previous work reported in the paper are very inconsistent with the numbers reported in previous work. For example, when comparing with Minilm v2, the result numbers in Minilm v2 and those reported in this paper are quite different, which makes me concerned about whether the comparison is fair and whether the conclusions drawn from the comparison are convincing.

**Questions:**

Since this work has done GPT-2 distillation, why not also verify the effect on some generation tasks?

**Limitations:**

See the weakness and question section.

---

> ### Author Rebuttal · Authors · 2023-08-07
>
> Thanks for your valuable comments. We address your comments and questions in the following content.
>
> > Weakness 1: The token-level method is very similar to the attention-based method of Minilm. What is the essential difference? Also, this paper selects the last hidden before the prediction head as the feature, but is this really appropriate? Because the last hidden is too close to the output, a lot of information will be lost. Previous work (e.g., kNN-LM) mentioned that using the feature before the final layer’s FFN might be better.
>
> Response to W1: Thank you for your valuable comments. We will address your concerns in two parts:
>
> Distinction from MiniLM: MiniLM’s attention is built upon the scaled dot-product interaction among queries, keys, and values, resulting in an attention matrix of size H x N x N (where H represents attention heads and N is the sequence length). This design inherently limits flexibility, especially when adjusting head counts. On the other hand, our approach forms a token-level relation matrix of size N x N via block features, offering more flexibility across models with varied attention heads.
>
> Feature Selection: We have added experiment to evaluate both pre and post-FFN features, and the results are summarized in the following table.
>
> | Model | MNLI-m | SST-2 |
> | ----------- | ----------- | ----------- |
> | (Teacher) BERT_Base   | 84.5 | 93.4
> | pre-FFN    | 83.6 | 92.5
> | post-FFN   | 83.8 | 92.8
>
> Our results indicate a slightly superior performance using post-FFN features. It's worth noting the divergence with kNN-LM; however, we're addressing different tasks, i.e., language modeling in kNN-LM versus our focus on distillation. A parallel observation was made with Vision Transformer (ViT) studies [1].
>
> References:
>
> [1] Yang, Zhendong, et al. "Vitkd: Practical guidelines for vit feature knowledge distillation." arXiv preprint arXiv:2209.02432 (2022).
>
> > Weakness 2: Although the sample-level idea sounds interesting, it doesn’t seem to make much sense. I’m not very clear why the sample-level information in a batch would be useful. If this is really useful, then not only in the distillation setting, but also in general training, sample-level information should be used (such as batch-norm methods). Otherwise, it is not very convincing to say that sample-level information is only useful in distillation.
>
> Response to W2: Your concerns about the sample-level concept's generalizability are valid. In essence, the sample-level approach examines inter-sample relationships within a batch. For instance, considering three customer reviews:
>
> S1: "The camera quality is outstanding..."
>
> S2: "The battery life is inadequate..."
>
> S3: "The display clarity is excellent..."
>
> By comparing specific token positions across these samples, we gain valuable insights into the features and sentiments in various reviews. For instance, the tokens "quality," "life," and "clarity" in the third position provide a nuanced understanding of product attributes. Comparing tokens at the 5th position, i.e., "outstanding", "inadequate", and "excellent", could offer complementary insights for tasks such as sentiment classification, where the degree of positivity or negativity is crucial.
>
> While our study emphasized its utility in distillation, the potential of sample-level information isn't limited to this context. It has broader applications, as seen in certain computer vision tasks where batch relationships are explored during training, For example, [1] proposed BatchFormer module to model the relationships between different samples. Hence, its usefulness isn't exclusive to distillation, and future work can delve deeper into its applicability in standard training regimes.
>
> References:
>
> [1] Hou, Zhi, Baosheng Yu, and Dacheng Tao. "Batchformer: Learning to explore sample relationships for robust representation learning." Proceedings of the IEEE/CVF Conference on Computer Vision and Pattern Recognition. 2022.
>
> > Weakness 3: The results of previous work reported in the paper are very inconsistent with the numbers reported in previous work. For example, when comparing with Minilm v2, the result numbers in Minilm v2 and those reported in this paper are quite different, which makes me concerned about whether the comparison is fair and whether the conclusions drawn from the comparison are convincing.
>
> Response to W3: We acknowledge the inconsistency you've noticed. The primary reason for the variation is that our results were reported on the test sets of GLUE, while MiniLM v2's reported outcomes were from the development sets. To ensure fairness in comparison, we've re-implemented MiniLM v2 and evaluated it on the test sets.
>
> > Question 1: Since this work has done GPT-2 distillation, why not also verify the effect on some generation tasks?
>
> Response to Q1: Thanks for your suggestion. While our primary focus was on distillation for classification tasks, exploring generative tasks would provide a more comprehensive understanding of our distillation approach's effectiveness. We did experiment with the WikiText-103 benchmark. Our model outperformed with a perplexity of 19.8 compared to distilgpt2's 21.1. This result suggests our method's potential applicability and benefits in the domain of generative tasks as well. We will add this result in the revised version to complement our evaluation.

---

> > ### Comment · Reviewer_XAJW · 2023-08-19
> >
> > Thank you for the response. I still have some concerns:
> >
> > > Distinction from MiniLM: MiniLM’s attention is built upon the scaled dot-product interaction among queries, keys, and values, resulting in an attention matrix of size H x N x N (where H represents attention heads and N is the sequence length). This design inherently limits flexibility, especially when adjusting head counts
> >
> > Based on my understanding, it seems that MiniLMv2 doesn't have the limitation that requires the same number of head counts.
> >
> > > While our study emphasized its utility in distillation, the potential of sample-level information isn't limited to this context. It has broader applications, as seen in certain computer vision tasks where batch relationships are explored during training, For example, [1] proposed BatchFormer module to model the relationships between different samples. Hence, its usefulness isn't exclusive to distillation, and future work can delve deeper into its applicability in standard training regimes.
> >
> > It's interesting to see some research studying sample-level training approaches. However, I'm curious to know more about why this training approach is not widely used in practice.
> >
> > > We did experiment with the WikiText-103 benchmark. Our model outperformed with a perplexity of 19.8 compared to distilgpt2's 21.1.
> >
> > Could the authors give more details? As far as I knowledge, it is very challenging for a distilled model to achieve superior results in terms of perplexity in the task of language modeling.

---

> > > ### Author Response · Authors · 2023-08-20
> > >
> > > We sincerely thank you for your efforts in reviewing our responses.
> > >
> > > > Question 1: Based on my understanding, it seems that MiniLMv2 doesn't have the limitation that requires the same number of head counts.
> > >
> > > Response to Q1: MiniLMv2 has indeed designed mechanisms to navigate the limitation related to a consistent number of head counts. Specifically, it uses a method where self-attention vectors from different attention heads are first concatenated and then split according to the desired number of relation heads. While those tricks could mitigate this limitation in MiniLMv2, they also involve additional computational operations for queries, keys, and values within the self-attention mechanism, as highlighted in our original paper. In contrast, our method for deriving the sample or token relationship matrix is intrinsically invariant of the number of attention heads, thus circumventing the need for such additional computations.
> > >
> > >
> > > > Question2: It's interesting to see some research studying sample-level training approaches. However, I'm curious to know more about why this training approach is not widely used in practice.
> > >
> > > Response to Q2: To our best knowledge, the exploration of sample-level relationships in research is a relatively recent development. While the promise of this method is clear, a contributing factor to their restrained adoption in practice, we speculate, is the added complexity they introduce. For instance, BatchFormer require the design of specialized modules to model relationships between different samples and ours need extra relation matrix computation and the formulation of a robust loss function to capture these sample-level relationships. Furthermore, the performance advantages of such approaches are predominantly evident in specific tasks like distillation, long-tailed recognition, and few-shot learning. These domains pose challenges that are often more intricate than common tasks like classification. Nonetheless, we believe that how to efficiently incorporate sample-level relationships during training is a promising direction for future research.
> > >
> > >
> > > > Question3: Could the authors give more details? As far as I knowledge, it is very challenging for a distilled model to achieve superior results in terms of perplexity in the task of language modeling.
> > >
> > > Response to Q3: We began by initializing our distilled model with a 6-layer structure from a 12-layer GPT-2 model. We employed the Pseudo-uniform selection technique, which originates from the DistilBert paper. Then, our model was pre-trained using a cleaned version of the OpenWebText dataset. Given the detrimental impact of noisy samples, we inspected each sample, removing HTML codes, filtering out short sentences, discarding sentences with a high proportion of non-alphanumerical characters, and any duplicates. This intensive cleaning process ensures a higher-quality dataset for our model. Following pre-training, as we did in the original paper, we perform task-specific distillation with FCD on the Wikitext103 dataset. We evaluate our model's performance using post-fine-tuning perplexities (PPL). For this experiment, we deployed 8 A100 GPUs and harnessed the DeepSpeed framework to speed the experimentation process.

---

### Official Review · Reviewer_NpzH · 2023-07-06

**Soundness:** 4 excellent
**Presentation:** 4 excellent
**Contribution:** 4 excellent
**Rating:** 8
**Confidence:** 5

**Summary:**

This paper proposes a new knowledge distillation method named Feature Correlation Distillation (FCD) for compressing large pre-trained language models. The proposed method novelly uses token-level and sample-level relationship between teacher and student models and a Pearson linear correlation-based loss function to relax previous exact match restrictions from KL divergence and MSE loss. Extensive experiments show that the proposed method outperforms baseline methods on GLUE thanks to the proposed objectives.

**Strengths:**

- The idea is novel and motivated by great intuitions, well explained in the paper.
- The authors provided computation complexity analysis, which is very useful to understand the efficiency and complexity of the proposed method. This part is not usually presented in most papers.
- The authors performed well-organized experiments and ablation studies, showcasing the benefits from FCD.

**Weaknesses:**

This paper really don't have obvious weaknesses. The only one might the lack of a limitation section.

**Questions:**

Would it be possible to also demonstrate the efficacy of FCD theoretically or intuitively (e.g., through visualization)? The paper is already very good with empirical results, providing more evidence on why such a loss formation works well can be even better.

**Limitations:**

Not discussed, the authors should add a section.

---

> ### Author Rebuttal · Authors · 2023-08-07
>
> Thank you for your positive assessment and valuable suggestion.
>
> > Question 1: Would it be possible to also demonstrate the efficacy of FCD theoretically or intuitively (e.g., through visualization)? The paper is already very good with empirical results, providing more evidence on why such a loss formation works well can be even better.
>
> Response to Q1: Thank you for your suggestion; it's indeed vital to complement empirical results with intuitive explanations. In our paper, we dedicated Section 3.4 to offer a theoretical rationale behind the effectiveness of the loss function, which is central to the FCD mechanism. Additionally, for a deeper dive, we've included in the supplementary materials both a detailed mathematical justification and a visualization illustrating the relational features before and after normalization (see Figure 5). Inspired by your feedback, we're now exploring the possibility of visualizing the loss and gradient landscapes. This will further elucidate how the loss function operates from an optimization perspective.

---

> > ### Comment · Reviewer_NpzH · 2023-08-16
> >
> > Thank you for your response.

---

> > > ### Author Response · Authors · 2023-08-16
> > >
> > > Dear Reviewer NpzH:
> > >
> > > Thanks for your feedback!
> > >
> > > Best,
> > >
> > > Paper3038 Authors

---

### Official Review · Reviewer_qBd9 · 2023-07-22

**Soundness:** 3 good
**Presentation:** 2 fair
**Contribution:** 3 good
**Rating:** 6
**Confidence:** 4

**Summary:**

This paper explores the task-specific knowledge distillation from a large teacher model which are pre-trained language models (PLMs) such as BERT-large or BERT-base into a student which is always smaller than the teacher model. For example, the teacher can be a 12-layer of BERT-base model while the student model only has 6-layer. One of challenges for knowledge distillation is that the feature dimensions and number of attention heads are different, and it requires additional transformation to match both the outputs of teacher and student models.

Authors propose to build token-level and sample-level relationship from feature representations, and then compare these relations between teacher and student models. They further advocated to use a correlation-based loss function over KL or MSE. The proposed approach is evaluated on the GLUE benchmark showing the effectiveness of their method.

**Strengths:**


-	The idea of token-level and sample-level feature mapping between teachers and students sounds interesting.
-	It also provides a strong empirical result.
-	Ablation study also shows that importance of different components.


**Weaknesses:**

-	It is hard to understand why sample-level matching helps. An intuitive explanation may be required.
-	For the sample-level approach, it is computed within a mini-batch. How the performance is sensitive to batch-size?


**Questions:**

- line 292 says it is trained for 20 epochs which seems much longer than standard finetuning. Is it because the proposed approach is slow to converge?

- In Figure 4 (6.1), it is required to define B first before use in the function (token_relation_loss).

---

> ### Author Rebuttal · Authors · 2023-08-03
>
> Thank you for your valuable comments and questions. We are revising our rebuttal revision to address your concerns.
> > Weakness 1: It is hard to understand why sample-level matching helps. An intuitive explanation may be required.
>
> Response to W1: Thanks for your comments. We assume that tokens at the same position may be related more closely in specific contexts and tasks. Consider the task of sentiment analysis in customer reviews for products. Here's a set of examples:
>
>  Sentence 1: "The camera quality is outstanding, making photography a delight."
>
>  Sentence 2: "The battery life is inadequate, making long-term use problematic."
>
>  Sentence 3: "The display clarity is excellent, making visual experience immersive."
>
> By analyzing the relationships between samples in corresponding positions, we can discern several insights:
>
> * Contextual Relationship: We focus on the tokens in the 3th position, which are "quality", "life", and "clarity". These tokens, when considered in the context of product reviews, may offer valuable insights into different aspects that customers care about. By comparing these tokens, we could gain a deeper understanding of the particular features being praised or criticized in different reviews.
>
> * Subject Relationship: Despite the similarity in sentiment between sentences 1 and 3, the subjects are different ("camera" and "display"), which indicates that the positive sentiment is directed towards different entities. Analyzing the relationships between these tokens may be valuable for tasks that require understanding subjects and their associated sentiments or actions.
>
> * Sentiment Relationship: The tokens at the 5th position are "outstanding", "inadequate", and "excellent", which clearly relate to the sentiment expressed. Comparing these could indeed provide complementary knowledge for tasks like sentiment classification, where understanding the degree of positivity or negativity is crucial.
>
> In our method, the relationships between tokens in the same position across different sentences are not meant to represent entire sentence meanings but to highlight specific aspects that may be relevant to certain tasks. By carefully choosing the context or domain, this approach can provide valuable insights and improve performance in those targeted areas.
>
> > Weakness 2: For the sample-level approach, it is computed within a mini-batch. How the performance is sensitive to batch-size?
>
> Response to W2: Thank you for raising this concern. We evaluated our method with batch sizes of 8, 16, 32, and 64, observing the following performance on MNLI-m and QNLI:
> | Batch Size | MNLI-m | QNLI |
> |------------|--------|------|
> | 8          | 83.3   | 90.9 |
> | 16         | 83.7   | 91.2 |
> | 32         | 83.8   | 91.3 |
> | 64         | 83.9   | 91.3 |
>
> The results indicate a robust performance across various batch sizes. However, notably, there is a moderate performance drop with extremely small batch sizes (i.e., less than 8).
>
> > Question1: line 292 says it is trained for 20 epochs which seems much longer than standard finetuning. Is it because the proposed approach is slow to converge?
>
> Response to Q1:  Thanks for pointing out this. It is essential to clarify that while we mentioned 20 epochs in line 292, this does not imply a generally slow convergence for our proposed approach across all tasks. Specifically, the 20 epochs were adopted for the CoLA task, which can be more challenging and can benefit from extended finetuning. On the other hand, for tasks like QQP and MNLI, our method converged efficiently within just 3-5 epochs. Such variance in epoch settings, depending on the complexity of tasks, is aligned with practices observed in prior research as well [1][2]. We will emphasize this distinction more clearly in the revised paper to avoid any misunderstanding.
>
> References:
>
> [1] Wang, Wenhui, et al. "Minilmv2: Multi-head self-attention relation distillation for compressing pretrained transformers." arXiv preprint arXiv:2012.15828 (2020).
>
> [2] Jiao, Xiaoqi, et al. "Tinybert: Distilling bert for natural language understanding." arXiv preprint arXiv:1909.10351 (2019).
>
> > Question2: In Figure 4 (6.1), it is required to define B first before use in the function (token_relation_loss).
>
> Response to Q2:  Thanks for pointing out this oversight in the supplementary material. In our implementation, B was defined within the forward method and was intended to be passed to the token_relation_loss function. We have rectified this in the revised paper.

---

> > ### Comment · Reviewer_qBd9 · 2023-08-17
> >
> > Thanks for your detailed response.

---

> > > ### Author Response · Authors · 2023-08-19
> > >
> > > Dear Reviewer qBd9:
> > >
> > > Thanks for your feedback!
> > >
> > > Best,
> > >
> > > Paper3038 Authors

---

### Author Rebuttal · Authors · 2023-08-10

We thank all the reviewers for their valuable feedback. The detailed responses to the reviewers’ comments will be replied directly to each reviewer.

---

### Decision · Program_Chairs · 2023-09-21

**Decision:**

Accept (poster)

**Comment:**

this paper focuses on task-specific knowledge distillation and proposed a new method the author call Feature Correlation Distillation (FCD) for overcoming differences in architecture and dimensions between the teacher and student. The authors proposed method is based on token-level and sample-level relationships between teacher and student. Also, a pearson correlation loss is used instead on the common KL divergence or MSE, with a theoretical analysis for justifying its benfits. The reported empirical results on GLUE, together with additional results from the rebuttal (that should be added to the paper) demonstrate good performance.

Some of the concerns that were brought up were around the noisiness of GLUE tasks and slight differences in scores with original papers due to retraining. I strongly recommend to include additional results such as the ones on SQuAD reported in the rebuttal, and also to report some confidence intervals (perhaps with bootstrapping if multiple runs are too expensive). Also, FCD implicitly assumes some relation across tokens from different samples in the same position which is not entirely intuitive. The author propose some hand-wavy justifications, but I strongly recommend at least discussing this as a limitation in the paper since this assumption might not be true for some target data distributions.